# *Dictyota* and *Canistrocarpus* Brazilian Brown Algae and Their Bioactive Diterpenes—A Review

**DOI:** 10.3390/md21090484

**Published:** 2023-09-04

**Authors:** Ana Débora Nunes Pinheiro Georgii, Valéria Laneuville Teixeira

**Affiliations:** Algamar Laboratory, Institute of Biology, Fluminense Federal University, Rua Prof. Frames Waldemar de Freitas Reis, Block M, São Domingos, Niterói 24210-201, RJ, Brazil; anadebora@gmail.com

**Keywords:** macroalgae, diterpenes, marine natural products, Dictyotaceae, biological properties

## Abstract

Dictyotaceae algae have gained recognition as prolific producers of diterpenes, which are molecules with significant biotechnological potential. These diterpenes hold immense promise as potential active drug components, making the algae a compelling area of study. The present review aims to present the latest advancements in understanding the biopotential of Brazilian *Dictyota* and *Canistrocarpus* brown algae, shedding light on the remarkable diversity and the biological and pharmacological potential of the secondary metabolites they produce. A total of 78 articles featuring 26 distinct diterpenes are reported in this review, with their antiviral potential being the most highlighted biological activity. Despite considerable research on these algae and their diterpenes, significant knowledge gaps persist. Consequently, the present review is poised to serve as a pivotal resource for researchers who are actively engaged in the pursuit of active diterpenes beyond the immediate purview. Furthermore, it holds the potential to catalyze an increase in research endeavors centered around these algal species within the geographical confines of the Brazilian coastline. Also, it assumes a critical role in directing future scientific explorations toward a better comprehension of these compounds and their ecological implications.

## 1. Introduction

Brown algae belonging to the Phaeophyceae class within the division Ochrophyta are predominantly found in marine environments and are typically attached to substrates. These macroalgae are not only the largest but also the most abundant benthic organisms in existence. Notably, two families, namely Sargassaceae and Dictyotaceae, contribute significantly to the vast repertoire of secondary metabolites discovered from brown algae. In fact, approximately 65% of all known secondary metabolites derived from brown algae can be attributed to these families. Their remarkable production of bioactive compounds highlights the potential of brown algae as a valuable source for various applications in diverse fields [1]. The economic interest in studying the Phaeophyceae class stems from the substantial number of molecules that have been reported from the secondary metabolism of this group of algae. In fact, research has identified more than 1140 molecules originating from the secondary metabolism of Phaeophyceae [1,2,3]. This wealth of bioactive compounds holds significant potential for various applications and industries. Researchers and industries alike are motivated to explore the Phaeophyceae class further in order to unlock its economic benefits [1,2,3,4].

The family Dictyotaceae plays a crucial role in the composition of benthic communities found in temperate and tropical seas, including the Brazilian coastline. With its extensive coastal belt, Brazil boasts a diverse marine ecosystem that encompasses a wide range of species, including brown macroalgae. Among these, representatives of the Dictyotaceae family are prominent components, contributing to the richness and ecological balance of the Brazilian coast [5].

The family Dictyotaceae, order Dictyotales, class Phaeophyceae, is composed of three genera: *Dictyota* Lamouroux, *Canistrocarpus* De Paula et de Clerck and *Rugulopteryx* De Clerck et Coppejans [6]. About 80 species of the Dictyotacea family were described, with 14 species in Brazil, although there are apparently more cited names than are properly recognized since some of them may be synonymous with each other [1,4,6,7,8,9,10,11,12,13,14,15]. The following species as taxa of *Dictyota* and *Canistrocarpus* are currently accepted in Brazil: *Dictyota bartayresiana* J.V. Lamouroux, *Dictyota ciliolata* O.G. Sond. ex Kütz, *Dictyota dolabellana* J.C. De Paula, Yoneshigue- Valentin and V.L. Teixeira, *Dictyota pfaffii* Schnetter, *Dictyota guineensis* (Kütz.) P. Crouan and H. Crouan, *Dictyota jamaicensis* H. R. Taylor, *Dictyota menstrualis* (Hoyt) Schnetter, Hörnig and Weber-Peukert, *Dictyota mertensii* (Martius) Kützing, *Dictyota pinnatifida* Kützing, *Dictyota pulchella* Hörnig and Schnetter, *Canistrocarpus cervicornis* (Kützing) De Paula and De Clerck, *Canistrocarpus crispatus* (J.V. Lamouroux) De Paula and De Clerck, *Dictyota caribaea* Hörnig and Schnetter, *Dictyota cuneata* Dickie, *Dictyota spiralis* Montagne, *Dictyota implexa* (Desf.) J.V. Lamouroux and *Dictyota crenulata* J. Agardh [7].

The objective of this review is to present a state-of-the-art assessment of the marine natural products from *Dictyota* and *Canistrocarpus* brown algae found along the Brazilian coast. More specifically, the focus of this assessment lies in exploring the diterpene compounds present within these algae, encompassing molecules that have been meticulously identified and characterized due to their notable biological (ecological) and pharmacological activities. A review of the activity of these diterpenes of brown algae of the genera *Dictyota* and *Canistrocarpus* may help researchers in the search for active diterpenes elsewhere in the world and the expansion of studies of these algae on the Brazilian coast. By delving into the activity profiles of these diterpenes sourced from *Dictyota* and *Canistrocarpus* brown algae, this review aims to offer valuable insights to researchers, which could potentially guide their endeavors in locating active diterpenes within other regions worldwide. Furthermore, this analysis holds the potential to catalyze the expansion of research efforts centered around these algae along the Brazilian coastline. The significance of this lies in enhancing our understanding of the diverse array of active compounds and their potential applications, ultimately contributing to broader scientific knowledge and facilitating the exploration of novel avenues in the realm of algal research.

This study also emphasizes the value of marine biodiversity as a reservoir of potential natural compounds possessing pharmacological properties or biotechnological prospects, which could be harnessed for the advancement of novel drug development.

## 2. Methodology

In order to carry out the review of Brazilian literature centered on marine natural products (diterpenes), a comprehensive search strategy was meticulously crafted to identify pertinent studies. Electronic databases, including PubMed and ScienceDirect, were pored using keywords such as “diterpenes”, “Brazil,” and “brown algae”. This thorough search yielded a wealth of over 200 articles. Subsequently, the reviewers performed an initial screening of titles and abstracts, painstakingly aligning them with predefined inclusion and exclusion criteria. The inclusion criteria encompassed Brazilian studies pertaining to diterpenes sourced from brown algae, while the exclusion criteria centered on reports of diterpenes lacking proven biological or pharmacological activity. Following this rigorous process, a total of 59 articles were carefully singled out for further analysis. In addition, the authors crossed the results with the marine natural product physical database curated by Professor Valéria Laneuville Teixeira. This database provides intricate details about each isolated diterpene, including information about its structure, collection site, taxonomic identifier, and associated biological activity(s). The review encompassed a specific subset of the database, focusing on diterpenes isolated from the Brazilian coast, with an emphasis on those with reported bioactivity, totaling 26 distinct diterpenes. To provide literary support, an additional 19 published studies was carefully chosen due to their significance in relation to the subject matter.

## 3. Dictyotaceae Diterpene Production

Marine organisms possess unique characteristics that set them apart from their terrestrial counterparts, primarily due to the need to adapt to extreme environmental variations in their aquatic habitats. These distinct features encompass various aspects, including behavior, metabolism, adaptation strategies, and information transfer mechanisms. These differences play an important role in shaping the diverse metabolic capabilities observed in marine organisms [10].

Dictyotaceae algae are known to be producers of molecules with special biotechnological potential, the diterpenes. The production of these substances is directly associated with the physiological responses from the algae to environmental factors, whether biotic (competition, herbivory, and microorganisms) or abiotic (temperature, salinity, light, and nutrients) [16].

*Dictyota* and *Canistrocarpus* are chemically composed of diterpenes [13,17,18,19]. These algae families are divided into three groups (I, II, and III), based on the biogenetic route of the diterpenes [17,20,21]. According to this proposal, the production of diterpenes occurs from the precursor geranyl-geraniol (Figure 1). Group I consist of metabolite-producing algae from the genus *Dictyota*, wherein the first ring cyclization takes place between positions 1 and 10, yielding diterpenes with prenylated derivatives of known sesquiterpene skeletal structure—the predominant framework within the *Dictyota* genus. Group II encompasses algae with diterpenes undergoing cyclization between positions 1 and 11. Group III comprises algae with diterpenes undergoing cyclization between carbons 2 and 10 of the precursor or exhibiting an intermediate formation akin to group I diterpenes, resulting in the contraction of the 10-carbon ring to 9 carbons. This gives rise to diterpenes with a xeniane skeletal structure [18].

Examples of species producing diterpenes of group I and III are *D. menstrualis*, *D. pinnatifida*, *D. crenulata*, and *D. ciliolata*. The division of group II into IIa and IIb was proposed by Bemfica et al. [9]. This group of researchers suggested that the algae producing dolabellanes belong to group IIa, while the group of dolastanes and secodolastanes and without the presence of dolabellanes are the group known as IIb, being both from the genus *Canistrocarpus* [8].

The differentiation between groups IIa and IIb lies in several key factors, including the cyclization pattern, stereoisomerism, and the presence of unsaturations. These distinctions ultimately result in the formation of diterpenes with structurally diverse skeletons.

The cyclization pattern refers to the specific arrangement of carbon atoms within the diterpene molecule, forming ring structures. Group IIa and IIb diterpenes exhibit different cyclization patterns, leading to distinct molecular frameworks and shapes. It is noteworthy that groups I, II, and III have representatives on the Brazilian coast.

The chemistry of algal diterpenes has contributed to the establishment of boundaries between species. As the identification and characterization of diterpenes from algae progress, it has become evident that these compounds can aid in defining the separation limits between genera, species, and even varieties. This is particularly important because taxonomists often face challenges in accurately classifying and categorizing algae based solely on morphological characteristics.

By studying the chemical profiles of algal diterpenes, researchers have identified specific metabolites that are indicative of certain taxonomic groups. These metabolites can be used as valuable markers to distinguish and differentiate taxa. The presence or absence of specific diterpenes, as well as variations in their chemical structures, can provide insights into the taxonomic relationships among algae [23,24,25].

Indeed, the terpene group, including diterpenes found in algae, is recognized for its potential biological and pharmacological activities. Diterpenes of Brazilian brown macroalgae are promising candidates for active drug principles, since many of them have proven pharmacological action, such as antivirals [26,27,28], inhibition of ATPase [29], antitumor, antibacterial, and antifungal. In addition, they act in herbivory relations [30].

## 4. Biopharmacological Importance of Diterpenes Produced by *Dictyota* and *Canistrocarpus* Brown Algae

The diterpenes derived from brown algae belonging to the Dictyoteae tribe have been the subject of extensive chemical investigations. These studies have resulted in the isolation and identification of over 350 diterpenes. These diterpenes have been discovered across a wide range of more than 30 different skeletal classes, highlighting the structural diversity within this group of compounds. The exploration of brown algae diterpenes has been conducted in various species found in oceans worldwide. More specifically, these investigations have encompassed around 20 different species distributed throughout different oceanic regions [16]. These metabolites have demonstrated remarkable in vivo activity and have been shown to participate in various important biological processes, performing different and important functions in the marine environment.

In addition, several studies have demonstrated that these substances have a role in chemical defense systems against herbivorous beings, resistance to biological incrustation, and allelopathic activity [31,32,33,34,35,36,37,38,39,40,41,42,43,44]. Many articles on the biological activities of diterpenes produced by algae of the Dictyoteae tribe have already been published. These studies include those with antitumor, antibacterial, and antifungal substances [45,46,47].

While most studies involving the identification and isolation of diterpenes have traditionally focused on algae collected from their natural habitats, recent research has expanded to include the cultivation of seaweed in laboratory settings [48]. Laboratory-based studies have successfully led to the identification of diterpenes in *D. menstrualis* [49]. The findings from the study revealed that populations of *D. menstrualis*, collected from various locations and time periods and subjected to technical replication, were able to survive for a duration of 4 to 6 months under laboratory conditions. Moreover, this study represented the first instance of detection of these metabolites under laboratory conditions [49]. These discoveries suggest that *D. menstrualis* could serve as a focal species for sustaining algal biomass, potentially offering a promising and sustainable approach for diterpene extraction.

The isolation and characterization of these diterpenes from various species of brown algae highlight the rich chemical diversity present in Brazil’s coastal regions. The testing of these molecules for their biopotential, which refers to their biological and pharmacological activities, provides valuable insights into their potential applications in various fields, such as medicine, biotechnology, and natural product research.

## 5. Biopotential of Brazilian Brown Algae of Groups I and III

*Dictyota menstrualis* (Hoyt) Schnetter, Hörnig, and Weber-Peukert is a brown alga characterized by the production of diterpenes from groups I and III of the biogenetic pathways. It exhibits a widespread geographical distribution, with a particular emphasis on the Americas and Caribbean islands. Within this alga, various diterpenes have been isolated, chemically characterized, and extensively studied for their biological properties. Numerous diterpenes derived from *D. menstrualis* have been identified and analyzed. These studies have shed light on their chemical structures and provided insights into their potential biological activities. One notable finding is that *D. menstrualis* possesses secondary metabolites with anti-HIV (human immunodeficiency virus) and anti-HSV (herpes simplex virus) properties. These properties make the alga and its diterpenes particularly interesting in the context of antiviral research. The investigation of these metabolites has contributed to our understanding of their mechanisms of action and potential applications in the development of antiviral therapies [26,50].

*D. menstrualis* is a producer of pachydictyol A (**1**) and isopachydictyol A (**2**) (Figure 2), diterpene molecules that can be found in other species of algae of the genus *Dictyota*, belonging to groups I and III. Both molecules present antiplatelet and anticoagulant activities, and they still have an anti-leishmania effect [51,52].

In 2004, researchers discovered the anti-HIV activity of two diterpenes isolated from brown algae. The first diterpene is called 6-hydroxy-dichotoma-3,14-diene-1,17-dial (**3**) (Figure 3), while its acetylated form is known as 6-acetoxy-dichotoma-3,14-diene-1,17-dial (**4**) (Figure 3) [26]. A little later, it was observed that the non-acetylated form has a greater inhibitory effect on the HIV-1 reverse transcriptase [48]. Later, the anti-HSV-1 effect of this algae was studied in relation to the dichotomane isolated 6-hydroxy-dichotoma-3,14-diene-1,17-dial (**3**) [53].

A study focusing on evaluating the antiviral potential of metabolites derived from *D. menstrualis* against the Zika virus (ZIKV) assessed the ability of crude extracts, acetylated crude extracts, and their fractions to inhibit ZIKV replication. The study observed a strong synergistic effect when suboptimal dosages of ribavirin were combined with the metabolites. This combination resulted in complete inhibition of viral replication [53].

Apart from their anti-viral effects, methanolic extracts derived from *D. menstrualis* have demonstrated potent induction of apoptosis in human cervical adenocarcinoma cells. This finding suggests the presence of molecules within the extracts that have the potential to act as antitumor agents [54].

*Dictyota mertensii* (Martius) Kützing, a brown seaweed, has been reported to exclusively produce diterpenes from group I, specifically diterpenes with a guaiane prenylated skeleton. This indicates that the diterpenes isolated from *D. mertensii* possess a unique structural characteristic. Among the diterpenes isolated from *D. mertensii*, dictyol H (**5**) (Figure 4) has been studied for its anti-herbivory activity. This specific diterpene has demonstrated activity against herbivorous organisms, indicating its potential role in protecting the seaweed from grazing or feeding by marine herbivores [41]. In this same study, dictyol B acetate (**6**) (Figure 4) was isolated.

Furthermore, it was shown in a different study that the dichloromethane and methanolic extracts of *D. mertensii* exhibit leishmanicidal activity [55]. This suggests that these extracts contain at least one molecule and possibly a combination of molecules that contribute to this specific pharmacological effect. This finding indicates the presence of multiple diterpenes with potentially bioactive properties in this species.

*Dictyota crenulata* J. Agardh is a species that appears to have a limited distribution in the oceans, possibly confined to the Atlantic Ocean and the west coast of Central America. Despite its restricted range, this species serves as a remarkably rich source of diterpenes. A study conducted in Brazil focused on specimens collected at Praia dos Padres in Espírito Santo and identified eight diterpenes in the crude extract using 1H-NMR and GC-MS analysis: pachydictyol A (**1**), isopachydictyol A (**2**), dictyotadiol (**7**), dictyol C (**8**), dictyodial A (**9**), 4β-hydroxy-dictyodial-A (**10**), 4β-acetoxy-dictylodial-A (**11**), and acetoxycrenulidane (**12**) [19]. Figure 5 shows the structures of the diterpenes 7–12.

This same study showed that in this specimen the 4β-acetoxy-dictyodial A (**11**) was the major diterpene. The identification of these diterpenes provides valuable insights into the chemical composition and potential bioactivity of this algae.

*Dictyota ciliolate* O.G. Sond. Ex Kütz has been previously described [56] as a potential complex of algae together with *D. crenulate*. However, in Brazil, these two species of brown algae are recognized as distinct entities. Notably, a methanolic extract of *D. ciliolata* collected in Natal, Brazil, exhibited antitumor activity by inducing apoptosis in human adenocarcinoma cells [56]. In a study conducted on samples collected from Atol das Rocas, Salvador, and Angra dos Reis, several diterpenes were identified: dictyol B acetate (**6**), dictyol B (**13**) (Figure 6), dictyol C (**8**), dictyoxide (**14**) (Figure 6), isopachydictyol A (**2**), pachydictyol A (**1**), 4β-acetoxydictyodial A (**11**), and four crenulidane diterpenes [57].

The presence of these diterpenes highlights the chemical diversity within *D. ciliolate*. Each of these compounds has its own unique chemical structure and may exhibit different biological properties and potential pharmacological activities.

*Dictyota guineensis* (Kütz.) P. Crouan and H. Crouan is a brown alga that occurs in the Bahia littoral [58]. According to reports, the diterpenes produced by the metabolism of this alga, namely dictyoxide (**14**), dictyotadiol (**7**), pachydictyol A (**1**), and isopachydictyol A (**2**) are the same as those found in other species [59]. Therefore, they do not present a specific chemotaxonomic marker exclusive to *D. guineensis*. While these diterpenes may not serve as specific chemotaxonomic markers for *D. guineensis*, they contribute to the overall chemical profile and potential biological activities of the species. The identification and characterization of these diterpenes provide valuable insights into the secondary metabolites produced by *D. guineensis* and contribute to our understanding of the chemical ecology.

*Dictyota caribaea* Hörnig and Schnetter, another species within the genus *Dictyota*, has been found to contain diterpenes that are also found in other algae of the same genus. Specifically, the presence of dictyol B acetate (**6**), pachydictyol A (**1**), and isopachydictyol A (**2**) has been also reported in *D. caribaea* [60]. The identification of these diterpenes in *D. caribaea* highlights the chemical similarities and shared metabolites within the genus. While these compounds have been previously identified in other species of *Dictyota*, their presence in *D. caribaea* further contributes to our understanding of the chemical diversity within this group of brown algae.

## 6. Biopotential of Brazilian Brown Algae of Group II

Within group II diterpenes, two subgroups can be distinguished: IIa and Iib. Group Iia is characterized by the production of diterpenes with a dolabellane skeleton, while group Iib consists of diterpenes with dolastane and secodolastane skeletons. In Brazil, two main representatives of group Iia are *Dictyota dolabellane* De Paula, Yoneshigue-Valentin et Teixeira and *Dictyota pfaffii* Schnetter.

These species are known to produce diterpenes with the dolabellane skeleton. The dolabellane diterpenes are structurally distinct from other diterpenes and exhibit unique biological properties.

*D. dolabellane* is noteworthy as the first denticulate species in which a dolabellane diterpene was discovered. The identification of a dolabellane diterpene in this specific species marked an important milestone in the field of natural product research. The diterpene found in *D. dolabellane*, namely 4-hydroxy-7,8-epoxy-2-dolabellane, possesses a distinctive oxidative pattern rarely observed in algae [19,61].

Regarding *D. pfaffii*, studies conducted on this species have revealed its potential as a source of bioactive compounds with anti-viral properties. These investigations have focused on the identification and characterization of specific molecules present in the alga that exhibit inhibitory effects against HIV-1 and HSV-1 [26,53,62]. In a notable study, the anti-HSV-1 activity of both 10,18-diacetoxy-8-hydroxy-2,6-dolabelladiene (**15**) and 10-acetoxy-8,18-dihydroxy-2,6-dolabelladiene (**16**) was reported [26]. These compounds were isolated and characterized, as well as 8,10,18-trihydroxy-2,6-dolabelladiene or dolabelladienetriol, (**17**), from populations collected in Atol das Rocas. Figure 7 shows the structures of diterpenes 16, 17, and 18.

The isolated dolabelladiene was tested for anti-HIV activity, specifically targeting the reverse transcriptase enzyme of HIV-1. The study revealed promising results, indicating that the dolabelladiene exhibited an inhibitory effect against the activity of the reverse transcriptase of virus HIV-1 [63]. More recently, the diterpene dolabelladienetriol (**18**) has shown promising potential for the development of a new commercial drug to treat viral infections caused by HSV-1 [64].

The isolated dolabellane diterpene, specifically dolabelladienetriol (**17**), has shown promising results as a potential anti-viral template. Its effectiveness against viral infections, particularly HSV-1, suggests its potential for developing new antiviral drugs [53]. Also, the enantiomeric diterpenes (S)-10,18-diacetoxy-7-hydroxy-2,8(17)-dolabelladiene (**18**) and (R)-10,18-diacetoxy-7-hydroxy-2,8(17)-dolabelladiene (**19**) (Figure 8) exhibited potent anti-HIV-1 activities and low cytotoxic activity against tumor cells of MT-2 lymphocytes. These findings indicate that these diterpenes might be promising as anti-HIV-1 agents [65,66].

Another species producing group II diterpenes with proven antiviral activity is *Canistrocarpus cervicornis* (Kützing) De Paula and De Clerck, previously known as *Dictyota cervicornis*.

*C. cervicornis* is one of the species of brown algae most studied in Brazil as a seaweed-producing diterpene of group Iib [9]. The studies with this alga began with Kelecom and Teixeira in the 1980s, when the isolation and chemical characterization of dolastanes and secodolastanes of this species were first described [16]. This species has been of interest due to its reported antiviral activity [67,68,69].

The diterpenes produced by *C. cervicornis* have indeed shown diverse and promising bioactivities. Two dolastane-type diterpenes isolated from *C. cervicornis* demonstrated antiviral activity against HSV-1. The diterpenes 4-hydroxy-9,14-dihydroxydolasta-1(15),7-diene (**20**) and 4,7,14-trihydroxydolasta-1(15),8-diene (**21**) (Figure 9) exhibited inhibition of HSV-1 infection in Vero cells, suggesting their potential as antiviral drugs.

A dolastane diterpene exhibited activity against *Leishmania amazonensis*, the causative agent of leishmaniosis. The specific diterpene responsible for this activity was not mentioned, but its discovery suggests the potential of *C. cervicornis* as a source of natural compounds for the development of antileishmanial drugs.

Recently, several discoveries were made with diterpenes produced by the *C. cervicornis* species: two diterpenes of the dolastane type presented antiviral activity against HSV-1; a dolostane that presented activity against *Leishmania amazonensis*; another that neutralized the hemolytic and obstructive activities of the *Lachesis muta* venom and inhibited the activity of the Na^+^K^+^-ATPase enzyme; finally, two secodolastanes that neutralized the hemolytic, proteolytic, and obstructive activities of the *Lachesis muta* venom [28,70]. Two dolostanes, 4-hydroxy-9,14-dihydroxydolasta-1(15),7-diene (**20**) and 4,7,14-trihydroxydolasta-1(15),8-diene (**21**), were isolated, and both molecules inhibited HSV-1 infection in *Vero* cells [15], suggesting that such structures are promising as antiviral drugs.

Two diterpenes were isolated from *C. cervicornis* collected in Angra dos Reis: 4-acetoxy-9,14-dihydroxydolasta-1(15),7-diene (**22**) and 4α,7α-diacetoxy-14-hydroxydolasta-1(15),8-diene (**23**) (Figure 10) [29].

These diterpenes showed good activity in the inhibition of the Na^+^K^+^-ATPase enzyme. It was also observed that the structural differences between these two diterpenes altered the inhibitory and selective power in the enzyme in question, suggesting that perhaps they are interesting structures for thinking about models of new enzymatic inhibitors.

In relation to other biological and pharmacological activities, antifouling, herbivory protection, and the inhibition of developing zygotes were attributed to an isolated dolastane, 4α,7α-diacetoxy-14-hydroxydolasta-1(15),8-diene (**23**) [43,68,69]. Isolinearol (**24**) and linearol (**25**) also diterpenes isolated from *C. cervicornis*, exhibit antifouling activity and provide protection against herbivory. Furthermore, these compounds are recognized for their antiophidian properties [71] as well as the 4α-acetoxy-9β,14α-dihydroxydolasta-1(15),7-diene (**22**) [72]. Protection against herbivory appears to be an interesting feature and it has also been attributed to 4α-acetoxyamijidictyol (**26**) Figure 11 [73].

A study showed the antiproliferative effect of crude extracts of *C. cervicornis* against amastigote forms of *Leishmania amazonensis* [74], suggesting the potential of *C. cervicornis* as a source of natural compounds for the development of antileishmanial drugs. In the mentioned study, it was demonstrated that the diterpene 4α-acetoxy-9β,14α-dihydroxydolasta-1(15),7-diene (**22**), which was previously isolated in a different study, exhibited dose-dependent activity [29]. This means that the biological effects of the diterpene were observed to increase or decrease with varying concentrations of the compound. This dolastane promoted cytotoxicity with mitochondrial damage, suggesting that it is an interesting molecule for the development of a new drug to be used in the treatment of leishmaniosis. Moreover, this dolastane was reported as an antifouling agent [43]; the larvicidal action against *Aedes aegypti* for dichloromethane and methanolic extracts was also described [74,75].

## 7. Conclusions and Final Considerations

This literature review provides a comprehensive overview on *Dictyota* and *Canistrocarpus* diterpenes, shedding light on their notable accomplishments in showcasing biological or pharmacological activities. A particular emphasis is placed on the remarkable anti-viral properties exhibited by dolastanes sourced from *C. cervicornis* and dollabelanes originating from *Dictyota pfaffi*. Moreover, the review delves into the prospective antiviral capacities of specific algae extracts, with a significant focus on *Dictyota menstrualis*.

Despite the substantial body of scientific research dedicated to these algae, there persist considerable voids in our understanding. The synthesis of knowledge thus far has been instrumental in unraveling the potential of these compounds, yet notable gaps continue to exist, necessitating further exploration.

It is crucial to recognize the substantial pharmacological potential of these Brazilian algae populations. The diterpenes derived from these algae hold promise for various applications in the pharmaceutical and biotechnological industries. Their demonstrated biological activities, such as anti-viral, anti-tumor, and anti-inflammatory properties, underscore their potential therapeutic value.

In order to fully utilize the potential benefits of these algae and their bioactive compounds, future studies should prioritize sustainable exploitation practices. This involves developing methods to fully unlock the commercial and industrial opportunities offered by these valuable marine resources. To achieve this, there is a need for continued exploration of the chemical diversity present in these algae as well as the identification of novel compounds that may possess unique and valuable properties. Elucidating the mechanisms of action of these compounds is also essential for understanding their potential applications in various fields, such as medicine, agriculture, and biotechnology.

## Figures and Tables

**Figure 1 marinedrugs-21-00484-f001:**
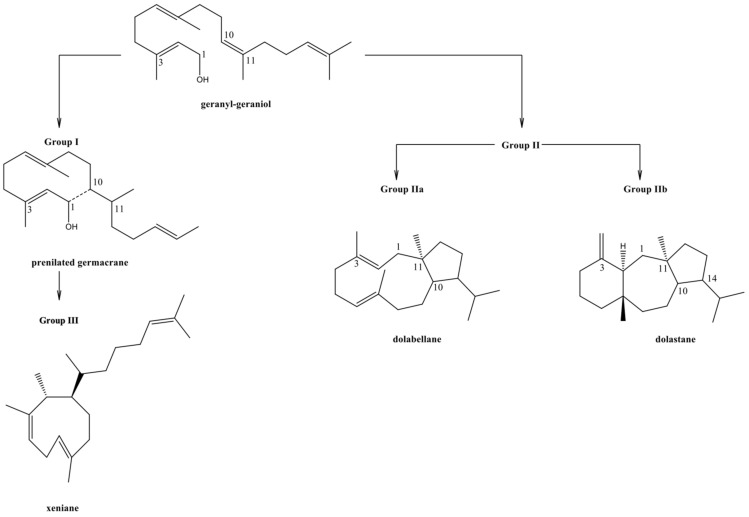
Diterpene skeletons of groups I, II, and III, according to the biosynthetic pathway [9,17,20,21,22].

**Figure 2 marinedrugs-21-00484-f002:**
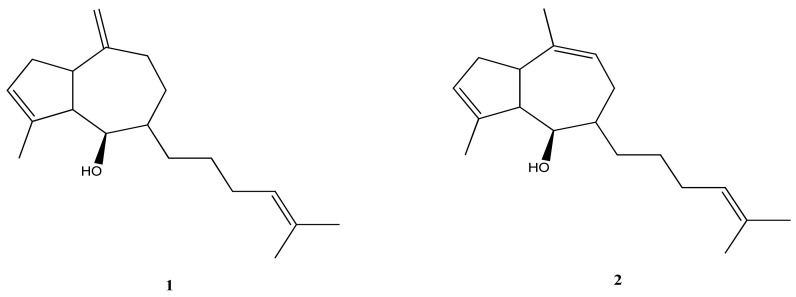
Structures of pachydictyol A (**1**) and isopachydictyol A (**2**).

**Figure 3 marinedrugs-21-00484-f003:**
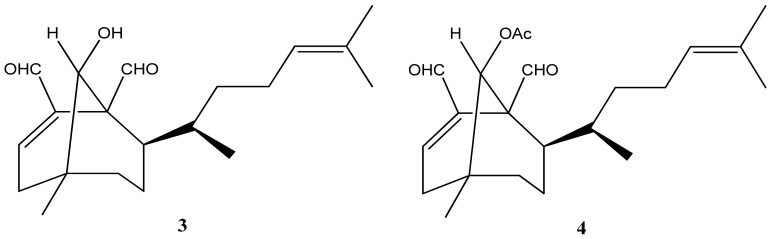
Structures of 6-hydroxy-dichotoma-3,14-diene-1,17-dial (**3**) and 6-acetoxy-dichotoma-3,14-diene-1,17-dial (**4**).

**Figure 4 marinedrugs-21-00484-f004:**
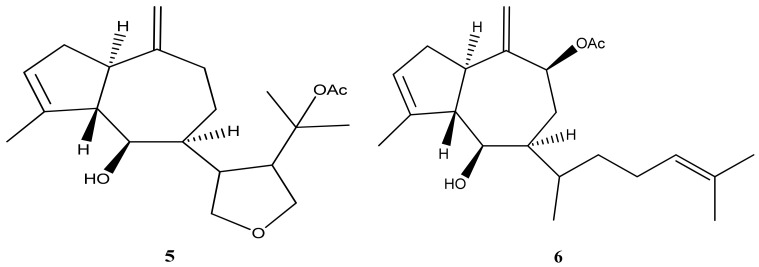
Structures of dictyol H (**5**) and dictyol B acetate (**6**).

**Figure 5 marinedrugs-21-00484-f005:**
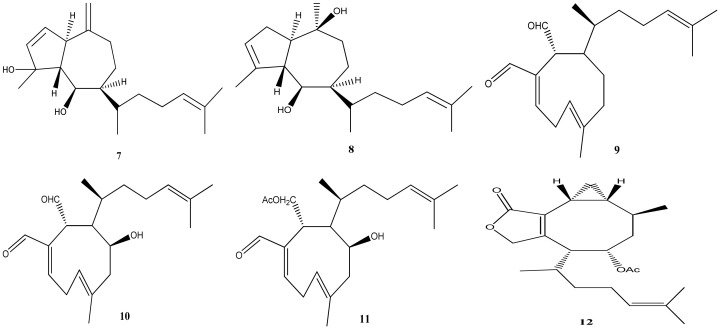
Structures of dictyotadiol (**7**), dictyol C (**8**), dictyodial A (**9**), 4β-hydroxy-dictyodial-A (**10**), 4β-acetoxy-dictylodial-A (**11**), and acetoxycrenulidane (**12**).

**Figure 6 marinedrugs-21-00484-f006:**
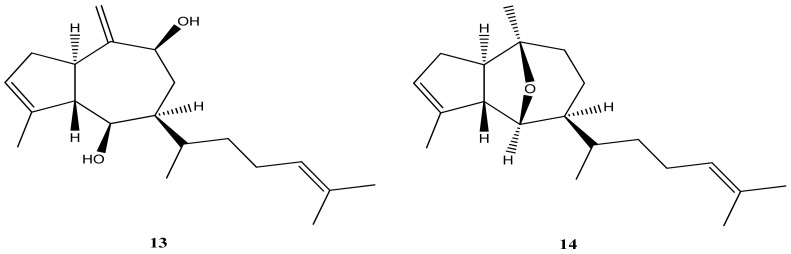
Structures of dictyol B (**13**) and dictyoxide (**14**).

**Figure 7 marinedrugs-21-00484-f007:**
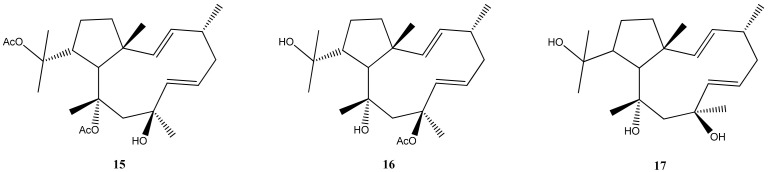
Structures of 10,18-diacetoxy-8-hydroxy-2,6-dolabelladiene (**15**), 10-acetoxy-8,18-dihydroxy-2,6-dolabelladiene (**16**) and 8,10,18-trihydroxy-2,6-dolabelladiene (**17**).

**Figure 8 marinedrugs-21-00484-f008:**
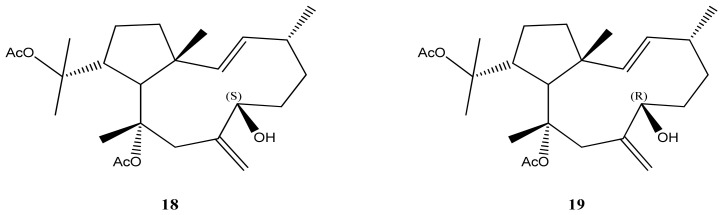
Structures of the enantiomeric diterpenes (S)-10,18-diacetoxy-7-hydroxy-2,8(17)-dolabelladiene (**18**) and (R)-10,18-diacetoxy-7-hydroxy-2,8(17)-dolabelladiene (**19**).

**Figure 9 marinedrugs-21-00484-f009:**
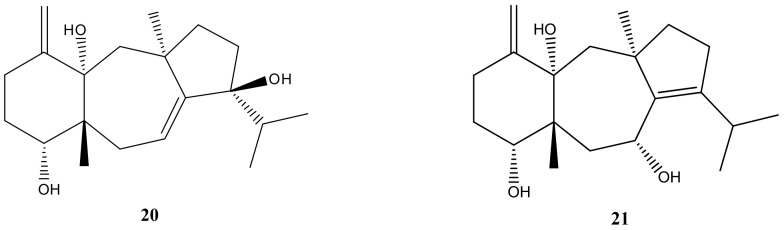
Structures of 4-hydroxy-9,14-dihydroxydolasta-1(15),7-diene (**20**) and 4,7,14-trihydroxydolasta-1(15),8-diene (**21**).

**Figure 10 marinedrugs-21-00484-f010:**
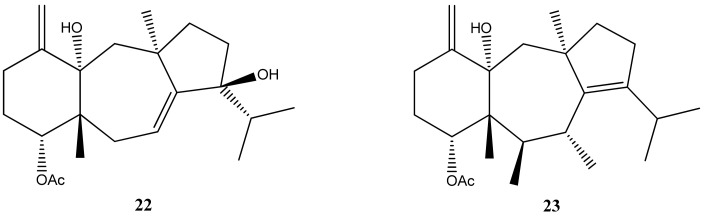
Structures of 4-acetoxy-9,14-dihydroxydolasta-1(15),7-diene (**22**) and 4α,7α-diacetoxy-14-hydroxydolasta-1(15),8-diene (**23**).

**Figure 11 marinedrugs-21-00484-f011:**
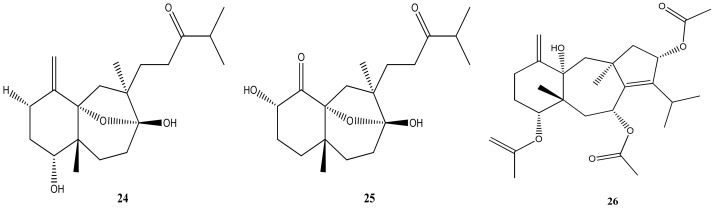
Structures of Isolinearol (**24**), linearol (**25**), and 4α-acetoxyamijidictyol (**26**).

## Data Availability

No new data were created or analyzed in this study. Data sharing is not applicable to this article.

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
