# Peer review of "Dictyota and Canistrocarpus Brazilian Brown Algae and Their Bioactive Diterpenes—A Review"

_marinedrugs, 2023, doi:10.3390/md21090484_

Round 1

Reviewer 1 Report

The manuscript is of interest and offers a worthy contribution to the field. However, some major revisions are required. The following comments should help further improve the quality of the work:

1. The author is careless in writing. The second paragraph is a repetition of the first paragraph. The English language used throughout the manuscript needs minor improvement as there are few punctuations and grammatical mistakes present.

2. The novelty/originality of the paper in the introduction section should be more effectively established.

3. The overall structure needs to be improved. Many sentences are easily cause misunderstanding.

4. Line 92-Line 94 “Dictyota and Canistrocarpus are chemically composed of diterpenes. Those 92 algae family were divided into three groups (I, II and III), based on the biogenetic route of 93 diterpenes.”Here, the authors are encouraged to describe the basis of the classification.

5. Line 120- Line 122:The authors emphasized the contribution of algal diterpenes to the classification of algae. However, what is the principle of diterpenes used for algae classification? Please supply the principle and the classification cases. “The presence or absence of particular diterpenes, as well as variations in their chemical structures, can provide insights into the taxonomic relationships among algae”. This part needs to be introduced in depth.

6. Kindly revise reference format according to the author guideline.

7. It is suggested to cite references within 5 years of research to maintain the reliability of results obtained.

The English language used throughout the manuscript needs minor improvement as there are few punctuations and grammatical mistakes present.

Author Response

REVIEWER 1

The manuscript is of interest and offers a worthy contribution to the field. However, some major revisions are required. The following comments should help further improve the quality of the work:

  1. The author is careless in writing.The second paragraph is a repetition of the first paragraph. 

The second paragraph of the introduction has been deleted, as recommended

The English language used throughout the manuscript needs minor improvement as there are few punctuations and grammatical mistakes present.

The revision time was insufficient for the English edition by specialized reviewers. We did a review carefully.

  1. The novelty/originality of the paper in the introduction section should be more effectively established.

We have modified and expanded the objectives of the review.

  1. The overall structure needs to be improved. Many sentences are easily cause misunderstanding.

We perform a revision and modify the text for better understanding.

  1. Line 92-Line 94 “Dictyota and Canistrocarpus are chemically composed of diterpenes. Those 92 algae family were divided into three groups (I, II and III), based on the biogenetic route of 93 diterpenes.”Here, the authors are encouraged to describe the basis of the classification. 

The basis of the formation (biogenetic proposals) of diterpenes is presented in the text of lines 137-145, the text of which has been revised.

  1. Line 120- Line 122:The authors emphasized the contribution of algal diterpenes to the classification of algae. However, what is the principle of diterpenes used for algae classification? Please supply the principle and the classification cases. “The presence or absence of particular diterpenes, as well as variations in their chemical structures, can provide insights into the taxonomic relationships among algae”. This part needs to be introduced in depth.

The authors believe that this suggestion is contemplated with the modifications made in the previous item 4.

  1. Kindly revise reference format according to the author guideline.

The authors performed a review of the references

  1. It is suggested to cite references within 5 years of research to maintain the reliability of results obtained.

The QN of Dictyota and Canistrocarpus n Brazil is much older than in the last 5 years and the revision would be very distorted from reality.

Reviewer 2 Report

This review highlights the biopotential of Dictyota and Canistrocarpus Brazillian brown algae diterpenes. This review is very well written.    

However, there are some minor points that require attention from author as below:

1.      I guess the abstract could be up to 200 words, according to MDPI. Please provide more informative information in the abstract. It will be good if author can add more background and purpose of this study in the abstract so reader can understand more about the importance of this review.

2.      Line 18: There is a repetition of “Keywords:”, please remove one.

3.      Paraghraph 1 and 2 are same, please remove one.

4.      Line 51: Change the reference number into 1-4.

5.      Line 74: One paraghraph must consist of more than one sentence. I am wondering why author should mention seaweed in this paraghraph, I think there is no cerrelation between this paraghraph with paraghraph before.

6.      Line 85: there is a typo “play na important” change into “an important”

7.      It is better for author to add conclusion part before final consideration part to make it easier for readers to understand the contents of this review as a whole.

Author Response

REVIEWER 2

This review highlights the biopotential of Dictyota and Canistrocarpus Brazillian brown algae diterpenes. This review is very well written.     

However, there are some minor points that require attention from author as below: 

  1. 1.I guess the abstract could be up to 200 words, according to MDPI. Please provide more informative information in the abstract. It will be good if author can add more background and purpose of this study in the abstract so reader can understand more about the importance of this review.

The abstract has been rewritten, as suggested by the reviewer, and now consists of 184 words.

  1. Line 18: There is a repetition of “Keywords:”, please remove one.

The authors made the suggested modifications.

  1. Paraghraph 1 and 2 are same, please remove one.

The authors made the suggested modifications.

  1. Line 51: Change the reference number into 1-4.

The authors made the suggested modifications.

  1. Line 74: One paraghraph must consist of more than one sentence. I am wondering why author should mention seaweed in this paraghraph, I think there is no cerrelation between this paraghraph with paraghraph before.

The authors made the suggested modifications.

  1. Line 85: there is a typo “play na important” change into “an important”

The authors made the suggested modifications.

  1. It is better for author to add conclusion part before final consideration part to make it easier for readers to understand the contents of this review as a whole.

The authors renamed it "Conclusions and final considerations", with a summary of what was discussed throughout the review in the first two paragraphs.

Reviewer 3 Report

The authors conducted a literature review on diterpenes from Dictyota and Canistrocarpus Brazilian brown algae and their bioactive potential, having importance in biotechnological development of these marine resources. However, I have the following observations:

Major:

o   Literature review should follow the PRISMA flowchart and include appropriate methodology that may include time duration, number of reference articles, sources of information, exclusion criteria and so on.

o   Introduction is inadequately written. The purpose of this review is not defined well.

o   Pharmacological effects of diterpenes should be categorized in a table to make the literature review reader-friendly.

Minor:

o   Line 22-36 and Line 26-51 are simply repitations in ‘Introduction’ part.

o   Line 36: same reference came twice, eg. [4][1–3][4]

o   Line 76: The objective of this review was to present

o   In figure 1, group III diterpene is missing.

Moderate improvement is required.

Author Response

REVIEWER 3

The authors conducted a literature review on diterpenes from Dictyota and Canistrocarpus Brazilian brown algae and their bioactive potential, having importance in biotechnological development of these marine resources. However, I have the following observations:

Major:

o   Literature review should follow the PRISMA flowchart and include appropriate methodology that may include time duration, number of reference articles, sources of information, exclusion criteria and so on.

The authors included the item Methodology to better explain how the data were obtained for the review.

o   Introduction is inadequately written. The purpose of this review is not defined well.

The objectives of the review have been rewritten, being included between lines 63 and 82.

o   Pharmacological effects of diterpenes should be categorized in a table to make the literature review reader-friendly.

The table would be very long, with a large number of pages. The authors preferred the presentation in text format.

Minor:

o   Line 22-36 and Line 26-51 are simply repitations in ‘Introduction’ part.

The authors made the suggested modifications.

o   Line 36: same reference came twice, eg. [4][1–3][4]

The authors made the suggested modifications.

o   Line 76: The objective of this review was to present 

The authors made the suggested modifications.

o   In figure 1, group III diterpene is missing.

The authors made the suggested modifications.

Round 2

Reviewer 1 Report

The authors have addressed most of the comments; they have also tried to make changes according to the reviewers' suggestions. After revisions, the quality of the manuscript has been adequately enhanced. Therefore, the manuscript could be considered for publication in the Journal.

Author Response

Thank You very much by the suggestions

Reviewer 3 Report

I found the author addressed the comments. However, regarding methodology, the information is still not clear. 'Inclusion criteria' is not defined. Auhtor did not follow a structured protocol like PRISMA for the screening of literature. The online link for the database is not included.

Minor modification is appreciated.

Author Response

Dear reviewer 3,

One of the authors (VLT) has experience with the natural products of Dictyota and Canistrocarpus of more than forty-five years, and has therefore generated a database over these years. The personal database was made with the articles published on diterpenes. The personal database has been made with all natural algae products articles belonging to the genera Dictyota and Canistrocarpus (including the ancient genera Dilophus, Glossophora and Pachydictyon) since 1970, and includes international and regional articles.The survey of the literature carried out is much more comprehensive than those data obtained in the Web of Science, etc.  We do not use the Prisma protocol (it is not mandatory). There's no way we're going to do that right now., as suggested by reviewer 3, because we had more complete information previously selected for this review. FWe made changes and modifications to make the methodology clearer, as suggested by reviewer 3.( text in yellow).

Thank you very much by suggestions.

Valéria Laneuville Teixeira